# Indigenous Community Fishing Practices in Nagaland, Eastern Indian Himalayas

**Etsoshan Y. Ovung [1],* , Lizabeni M. Kithan [2], Francis Q. Brearley [3] and Shri Kant Tripathi [1]**

1 Department of Forestry, School of Earth Sciences and Natural Resource Management, Mizoram University, Aizawl 796004, India; sk_tripathi@rediffmail.com
2 Department of Agronomy, Tamil Nadu Agricultural University, Coimbatore 641003, India; liz.kits@gmail.com
3 Department of Natural Sciences, Manchester Metropolitan University, Manchester M1 5GD, UK; f.q.brearley@mmu.ac.uk
* Correspondence: etso123ovung@gmail.com

**Abstract:** The significance of indigenous knowledge under the current scenario of biodiversity imperilment is well-known since such knowledge is gained through continuous intergenerational observations of natural systems. In this study, we present a description of indigenous community fishing practices in Nagaland and investigate their relationship with the cultural and traditional aspects of the associated communities through oral interactions, questionnaires and as a participatory observer. We observed inter- and intra-community fishing in which the piscicidal plants *Millettia pachycarpa* and *Derris elliptica* (both Fabaceae) were used as fish poison. *M. pachycarpa* was commonly used in inter-community 'fishing festivals' since it is easily available, less laborious to collect and there are no reports of harm to the body in comparison to *D. elliptica* which causes allergy and/or dermatological effects. Indigenous community fishing is conducted to develop a sense of peaceful co-existence and prosperity within and among the neighboring communities. However, the increasing use of synthetic fish poisons has overlapped with the traditional practices of fishing, exerting pressure on the livelihoods and food security of the tribal populations while contributing to riverine ecosystem degradation. Formulation of policies banning synthetic fish poison, and judicious use of traditional piscicidal plant fishing is therefore recommended.

**Keywords:** community fishing; piscicidal plants; sustainability; traditional knowledge





## 1. Introduction

In the current era of increasing anthropogenic pressure on global biodiversity, there is an urgent need to understand the various human–nature relationships and identify the viable indigenous knowledge and practices that can augment the bio-geophysical systems [1]. The need for, and importance of, indigenous knowledge and practices has gained recognition in relation to conservation and sustainable resource management [2–4]. Promotion of this indigenous knowledge and practices can enhance effectiveness, sustainability and legitimacy in the management of ecosystems [5]. Indigenous fishing using piscicidal plants is one age-old traditional practice that has been transmitted intergenerationally and practiced to date and is significant with respect to environmental, cultural and livelihood sustainability. Fish poisons of plant origin are used for cultural, commercial and environmental reasons in waterway management for the control of non-game fish species [6]. Many plants contain chemicals which have traditionally been used to harvest fish and also to monitor various pests in almost all parts of the world [7]. Furthermore, the significance of local and traditional knowledge of fishing practices is well known globally with its importance being reflected in several studies involving the application and incorporation of the knowledge of the elders [8,9].

Fishing practices involving plant-based poisons (ichthyotoxins or piscicides) for cultural, commercial and environmental purposes are observed in many parts of the

world [10–13]. In India, traditional fishing practices are common among the various tribal groups wherein numerous plant species are utilized [14]. Indigenous methods of fishing are reported from various parts of Northeast India including Arunachal Pradesh, Assam, Manipur and Meghalaya [15–18]. Over 300 plant species with piscicidal properties have been reported from India [14], with over 100 of these reported from the northeastern region [6]. Nagaland is one of the states of Northeast India and hosts a rich diversity of fish species. A study from the area reported 197 fish species belonging to 10 orders, 26 families and 87 genera [7] within 1600 km of the major rivers [19]; however, documentation of traditional fishing activities in the state is scarce. The practice is conducted with varying levels of social complexity: smaller within tribe activities as well as larger complex between tribe 'fishing festivals' are carried out. This practice is of great indigenous value in its contribution to preserving and sustaining traditional knowledge for future generations. Indigenous community fishing has been a part of the cultural heritage of the Naga tribal populations, yet documentation, literature or records relating to this practice are scanty. Thus, it is important to further study the traditional methods of fishing to understand the procedural events and the cultural importance related to the practice(s) [3]. Furthermore, the impacts of such traditional practices on the environment are also important facets with respect to conservation and sustainability under the current scenario of rapid environmental degradation and biodiversity loss. Therefore, this study was initiated to explore the indigenous methods of piscicidal fishing in Nagaland to provide a detailed systematic procedure of the fishing activities while attempting to investigate their relationship with the cultural and traditional aspects of the associated communities.

## 2. Materials and Methods

### 2.1. Study Area

Northeast India is characterized by high forest cover, heavy monsoonal rainfall and mountainous topography with numerous valleys and rivers. One of the states of Northeast India is Nagaland with an area of 16,579 km$^2$ [20]. The economy of the state is based predominantly on agriculture, horticulture, animal husbandry, plantations and other forest products. The Nagas comprise more than 20 tribes and numerous sub-tribes with diverse languages and traditional cultures [21] and therefore there is complexity in the documentation of traditions. The majority of the rural tribal population is still dependent on traditional shifting (*jhum*) cultivation, hunting and fishing for their sustenance [22]. Under the current scenario of increasing human population leading to increased environmental pressure and biodiversity loss, some of these practices may no longer be sustainable. However, many of the old traditions and practices retain vital significance to avoid the erosion of customary traditions and practices due to the onset and impact of modern alternative practices.

This study was carried out along the Doyang and Jüthümrhü rivers flowing through Wokha and Zunheboto districts of Nagaland. The Doyang is the state's longest river originating from Japfu, Nagaland's second-highest peak, and running northward through Kohima, Zunheboto and Wokha districts before flowing into the Dhansiri and then the Brahmaputra in Assam. Apart from the Doyang Hydro Electric Project, the river is known for the availability of various fish species [23,24]. For this study, Chudi, Tsungiki and Mungya villages of the Lotha tribe and Philimi village of the Sumi tribe from Wokha and Zunheboto districts, respectively, were the study locations (Figure 1).

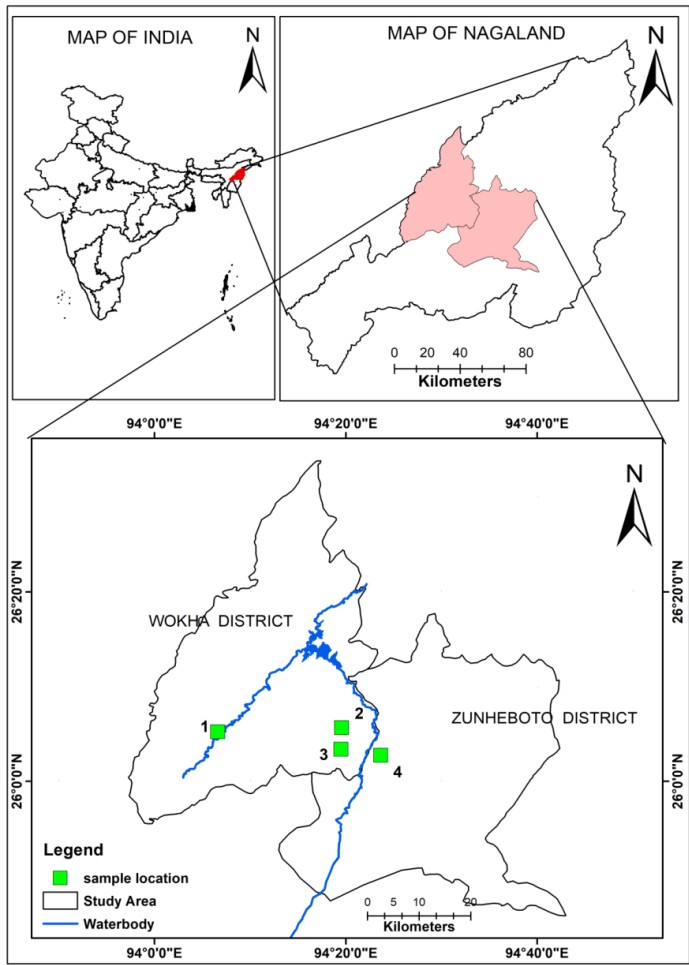

**Figure 1.** Map showing India, Nagaland and Wokha and Zunheboto districts with the four study villages (sample location): Chudi (1), Tsungiki (2) and Mungya (3) of Wokha district and Philimi (4) of Zunheboto district. The Waterbody is the Doyang River flowing north, through the Doyang Reservoir before flowing to the south; the tributary flowing from the north is the Chubi River.

### 2.2. Research Approach

This investigation was carried out based on data acquired through formal as well as informal oral interactions and interviews, questionnaires and as an independent participatory observer in situ in order to document the process starting from collection of the piscicidal plants till the harvesting of the fish. The study was carried out during the months of April and May 2021 on three different occasions out of which inter-tribal (Sumi and Lotha) community fishing was observed on two occasions while the intra-tribal (Lotha tribe only) activity was observed on one occasion (Table 1).

**Table 1.** Study villages and brief details on community fishing activities in Nagaland, Northeast India.

| Participating Tribe(s) and Village(s) Involved | Place of Activity (Local Names) | No. of Participants | Piscicidal Plant Used |
|---|---|---|---|
| Lotha (Tsungiki village) and Sumi (Philimi village) | Jüthümrhü | c. 3500 (male and female) | *Millettia pachycarpa* Benth. |
| Lotha (Mungya village) and Sumi (Philimi village) | Jüthümrhü | c. 500 (male and female) | *Millettia pachycarpa* Benth. |
| Lotha (Chudi village) | Pofü | 20 (male) | *Derris elliptica* (Wall.) Benth. |

Information was collected through oral interviews and interactions with the elders, tribal leaders and the participating rural community members. Twenty individuals from each tribe were also interviewed to obtain information, for example, on plants used, procedures involved, traditional heritage associated with the practices, etc. Out of the total number of interviewees who participated, 15 were elders, wherein 10 elders belonged to the Lotha tribe of the Wokha district and 5 elders were from the Sumi tribe of Zunheboto district. Subsequently, one tribal leader from each of the Lotha and Sumi tribes was also interviewed on separate occasions. The ages of the elders ranged from 65 to 80 years while the ages of the tribal leaders ranged from 50 to 60 years.

## 3. Results

In this study, different tribes from two districts, Wokha district (native district of the Lotha tribe) and Zunheboto district (native district of the Sumi tribe), were involved in the traditional fishing activity and two different types of practices, viz. inter- and intra-community fishing practice, were observed which differed in relation to their modes of operations, execution and number of participants and the plant used. We observed that the use of roots of the plant was the same in both practices even though the two species differed. Inter-tribal community fishing is also considered as a 'fishing festival' and therefore it involves a series of events and preparations a number of weeks prior to the date of the activity due to its complexity with large numbers of participants (Table 1). Community fishing between the Lotha and Sumi tribes was observed on two occasions where *Millettia pachycarpa* Benth. was used as the fish poison. While on another occasion, a different method of traditional fishing within the Lotha tribe was carried out using *Derris elliptica* (Wall.) Benth. (both Fabaceae). The following procedures involved in the activities are recorded, documented and described below.

### 3.1. Piscicidal Plants Used in Community Fishing

We found that *M. pachycarpa* (Figure 2a) locally called *Ono* or *Ngeri* in Lotha was the most commonly used piscicidal plant in community fishing. *M. pachycarpa* was used during the inter-tribal activities, while the intra-tribal activity used *D. elliptica* (Figure 2b) known as *Notsü* in Lotha, which shares a common family (Fabaceae) with the former species and is within one of the families in which piscicidal plants are most abundant [14]. Both *M. pachycarpa* and *D. elliptica* contain the compound rotenone ($C_{23}H_{22}O_6$) and are among the most commonly used plants for stupefying fish in the study area as well as more broadly in India [14]. Scientific studies state that the effect of piscicidal plants on fish is due to the inhibition of oxidative phosphorylation, a biochemical reaction which takes place in the energy-producing mitochondria within animal cells [25,26] leading to deoxyfication forcing the fish to come to the water surface. The impact of the applied piscicide differs among fish species [26] and the amount of poison used during a particular occasion is uncertain since it depends on a number of factors including the number of participants and the length, depth and volume of water in the river.

### 3.2. Procedure of Events during Community Fishing
#### 3.2.1. Initiation

Community fishing is usually carried out during April when the water level is low before the onset of monsoon rains. Initially, the village chief (*Gaon burah*), village elders and officials of both the villages of the two participating tribes call a meeting to discuss and plan events for the whole fishing process, while also fixing the ideal date for the venture. This usually happens weeks, or even months, beforehand. During this period, the plan of the operation and traditional customs which have to be followed are shared among the participating tribes, and the location is also decided. This study revealed that only men are allowed to participate in the activity right from the start of the operation. However, during the harvesting of the fish, women are then allowed to participate. Traditional fishing within the tribe (intra-tribal activity) is carried out under the direction of the *Gaon burah*, elders and

a chairperson. Thereafter, members of the community gather for a meeting to discuss the procedures and the elders share traditional values/customs related to the fishing practice.

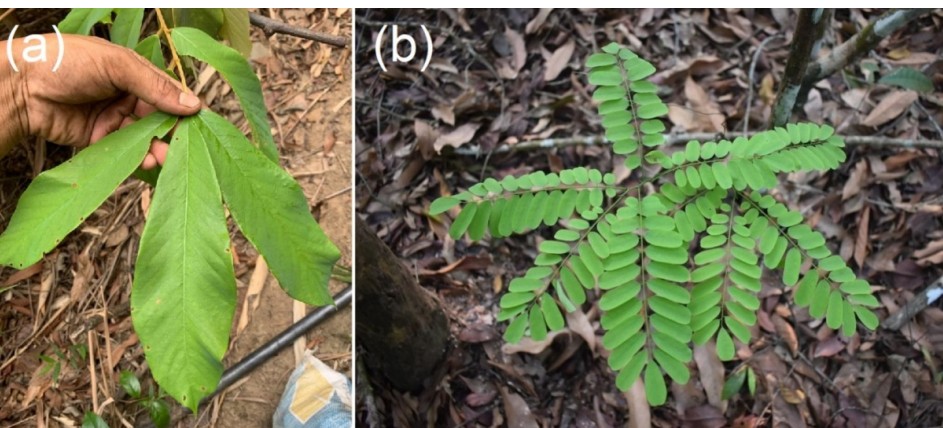

**Figure 2.** (**a**) *Millettia pachycarpa* and (**b**) *Derris elliptica* (both Fabaceae) plants growing in situ in Nagaland, Northeast India.

### 3.2.2. Collection of Plant Parts Used as Fish Poison

Prior to collection of the piscicidal plants, preliminary surveys are carried out to assess appropriate locations with availability of the plants to be extracted. This information is then shared among the participants. Men gather for a group discussion where the various participants are distributed into groups for extraction of the piscicidal plants. Collection of the roots of the two target species *M. pachycarpa* and *D. elliptica* is carried out a day prior to the proposed fishing date. Tools such as spades/shovels, machetes (*Dao*) and specially prepared spears are used for digging and extraction of the roots (Figure 3a). The collected roots are then cut into approximately equal lengths for further processing (Figure 3b,c). The roots are then stored in a single temporary depot and subsequently carried to the fishing location on the day of collection for further processing. Extraction and collection of the *M. pachycarpa* roots is reported to be less laborious relative to *D. elliptica* which requires more manpower and tools. This is likely linked to the shallower climbing plant root system of *M. pachycarpa* [27] in contrast to the roots of *D. elliptica* which are deeper making them harder to harvest.

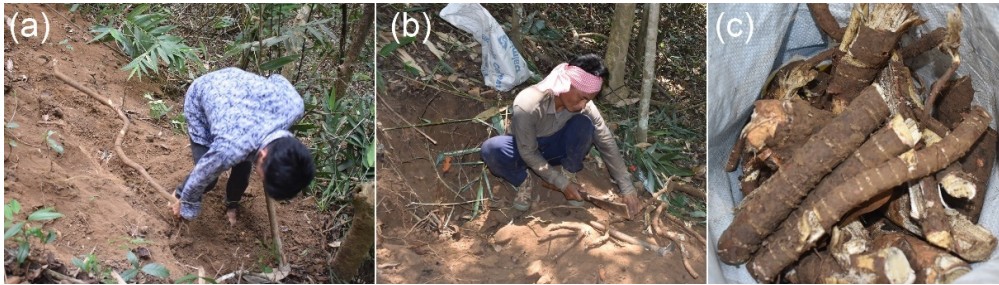

**Figure 3.** (**a**) Extraction and collection of roots and (**b**) cutting of roots into (**c**) equivalent sizes in preparation for community fishing in Nagaland, Northeast India.

### 3.2.3. Preliminary Pounding

On the same day after the collection of the roots, they are carried to the village or the fishing location. The collected roots are then cut into equivalent sizes, if not already completed, whereby *M. pachycarpa* roots are cut to lengths of about 30 cm while *D. elliptica* roots are cut into lengths of about 15 cm and pounded using a piece of wood (pole) against a stone or log (Figure 4) so as to separate the fibers and extract the piscicidal compound during the time of fishing. *M. pachycarpa* turns reddish in color after the preliminary pounding while *D. elliptica* roots exhibit a whitish color after pounding and produce a pungent smell. All the collected roots are pounded till the fibers of the plant become

separated. The roots are then bound into bunches using available natural ropes or strings made from young bamboos (*Bambusa* spp.; Poaceae). This practice is common for both the participating tribes, and each tribe is responsible for the collection, storage and use during the fishing event.

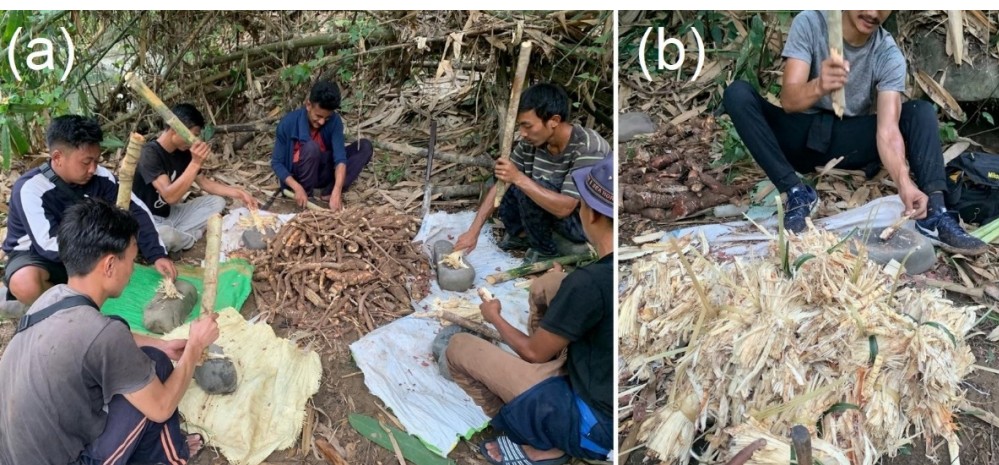

**Figure 4.** (**a**,**b**) Preliminary pounding of roots of *D. elliptica* prior to community fishing in Nagaland, Northeast India.

### 3.2.4. Production of Platforms for Pounding and Mass Pounding of the Piscicidal Roots Inter-Community Activities

This was observed during the two occasions of inter-tribal fishing between the Lotha and Sumi tribes, where each tribe constructs their own respective platforms. Prior to the second pounding of the bunches of roots for extraction and release of the fish poison, platforms are constructed by arranging them across the river using logs carried from the forest by the men (Figure 5a) to accommodate all those participating in the pounding of the roots. The length and width of the log platforms depend on the number of participants and the dimension of the river with sufficient gaps maintained between the logs for the participants for ease of pounding. The community members arrange the log platforms in such a manner that each community faces the other at a distance of about 25 m. Some logs are fastened together by placing smaller logs perpendicular to the platforms to avoid them rolling during the process (Figure 5b). In this inter-community fishing activity, *M. pachycarpa* roots are used as the piscicide. The preliminary pounded roots are arranged in bunches which are then brought to the riverside where the pounding platforms have been constructed. Only men are allowed to participate in the whole process of community fishing; during the two activities, we only observed men (including male children) participating in the pounding of the roots while the women waited on the banks until the harvesting of the fish. As a traditional practice, before pounding the roots, the oldest person present or the chairman initiates the activity invoking safety and luck through the exchange of speeches, prayers, cultural quotes and a war cry before starting the activity. A small tree pole is cut and used as a hammer for pounding the roots (Figure 6a) which is carried out for about 40 to 60 min. During this time, the bunches of roots are beaten continuously and frequently drenched in water to enhance the release of the chemical compounds. The large group of males participating from each tribe raise their pounding poles, singing and chanting war cries as a sign of respect to the ancestors and to each other, and initiate the pounding in an organized rhythmic manner with one community facing the other who take it in turns to pound their roots. The continual rhythmic pounding and chanting is an ancestral practice which signifies unity, respect and peaceful harmony (Figure 6b). The root juices produce a brown-reddish color (Figure 7) and the beaten roots are then dropped in the current of the water body to contribute to the efficacy of the poison. This particular event can be considered as one of the biggest inter-community fishing festivals in Nagaland and is integral to the preservation of cultural heritage of the two Naga tribes.

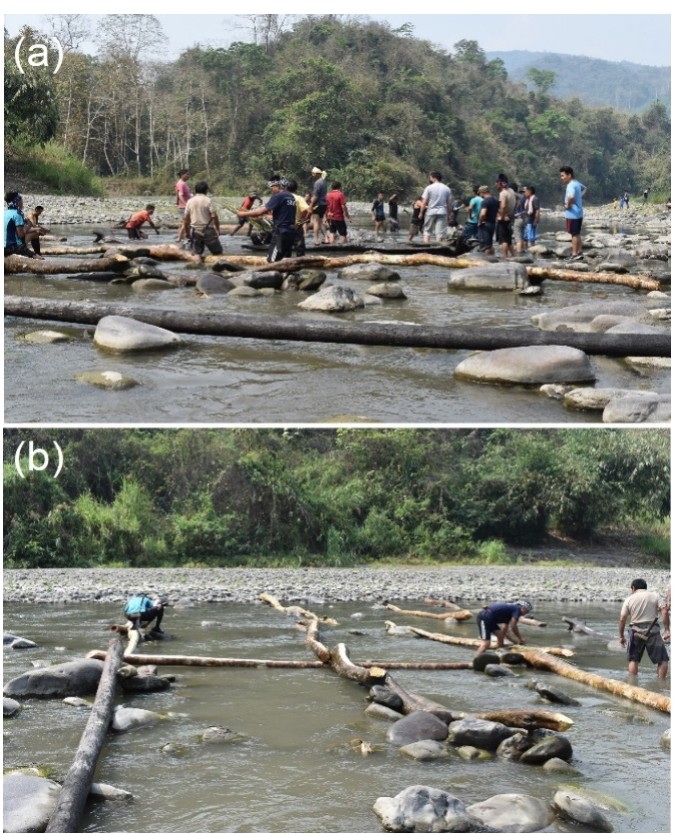

**Figure 5.** (**a**,**b**) Construction of wood platforms across the river for the pounding of *M. pachycarpa* for community fishing in Nagaland, Northeast India.

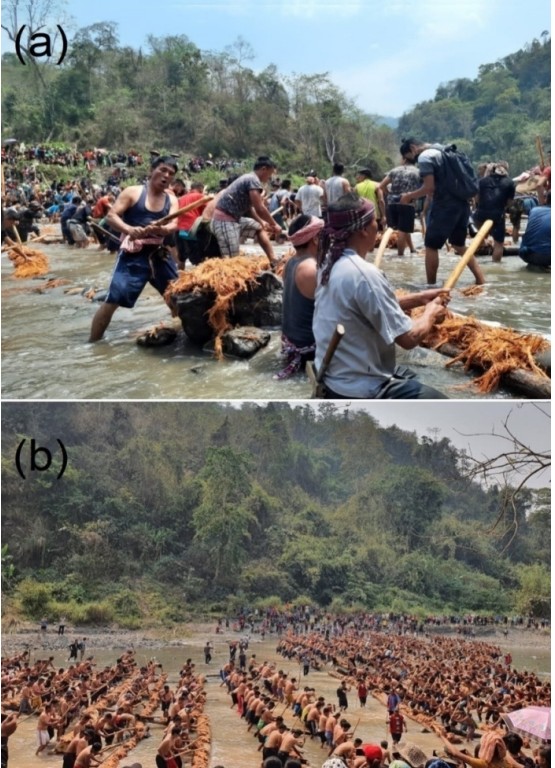

**Figure 6.** (**a**) Pounding of *M. pachycarpa* roots with short poles by (**b**) a large group of men from two villages in Nagaland, Northeast India.

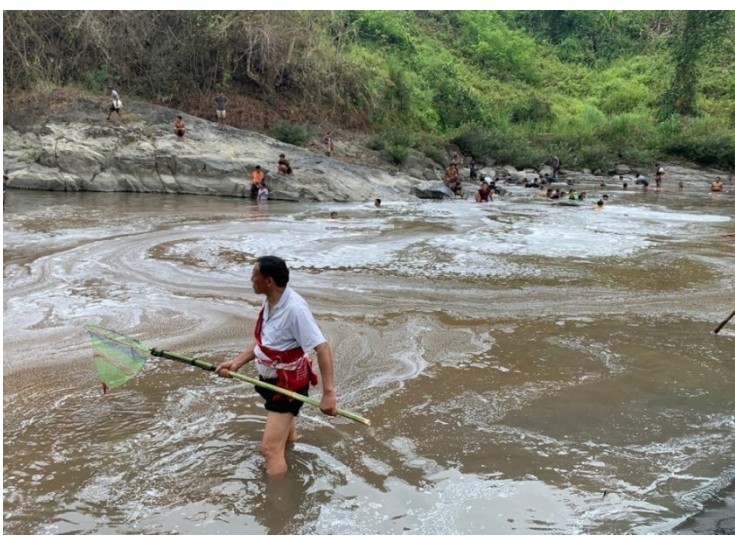

**Figure 7.** Brown-reddish extracts of *M. pachycarpa* produced after pounding for community fishing in Nagaland, Northeast India.

Intra-Community Activities

Another survey and observation of intra-community fishing was also carried out at Pofü in Chudi village of Wokha district. The process of plant root collection was similar to that already described; however, in this practice, the plant used was *D. elliptica.* Lotha villages such as Chudi, as well as Lotsu and Pyangsu of Wokha district, commonly use *D. elliptica* for community fishing. In this case of intra-community fishing, no women are allowed to participate in any of the activities from the beginning till the end of the activity as per traditional customs. However, the traditional pounding method in this practice differs from that of the inter-community fishing as described above. The platform constructed in the river for pounding is very different from the former as it is constructed by gathering stones at two or three places in the river (depending on the number of participants) where the piscicidal poison is to be pounded and released. Stones are gathered from across the river and arranged in a toroidal 'donut' shape (Figure 8a). The width of the outer arrangement of stones measures about 60 cm and serves as a platform for standing upon. Firstly, the preliminary pounded stored roots of *D. elliptica* are put in the circular void containing water (Figure 8b) and long poles are used as a tool for pounding the roots whilst standing upright on the outer platform (Figure 8c), which is then followed by the beating/threshing of the roots with shorter wooden poles. The reason for pounding the roots of *D. elliptica* whilst standing is suggested as a precautionary measure to reduce contact with the poison released during the pounding. Initial pounding of the plant produces a higher quality (i.e., greater concentration) of the chemical compound which may sometimes cause allergy and/or swelling of eyes and ears in cases of contact. Pounding of the roots in the void is usually carried out for about an hour. Furthermore, the pounded roots are beaten/threshed again so as to extract as much poisonous compound as possible for another 40 min. The remains of the roots are then released into the current of the water body (Figure 8d) to increase their efficacy. It was also revealed that the poison of *D. elliptica* travels through the water body in a linear pattern (Figure 8c) whereas the extracts of *M. pachycarpa* spread more evenly throughout the water body (Figure 7). The color released from the pounded root of *D. elliptica* is yellowish (Figure 8c). It was reported that *D. elliptica* has higher efficacy than *M. pachycarpa* related to the reports of negative effects from contact with *D. elliptica* as noted above. From interviews with the elders, it was traditionally believed that the toxicity of *D. elliptica* roots continues to affect the fish for up to nine days although *M. pachycarpa* affects the fish for only a couple of days which is in contrast with other scientific studies [28–30] which showed a fairly short ecological half-life.

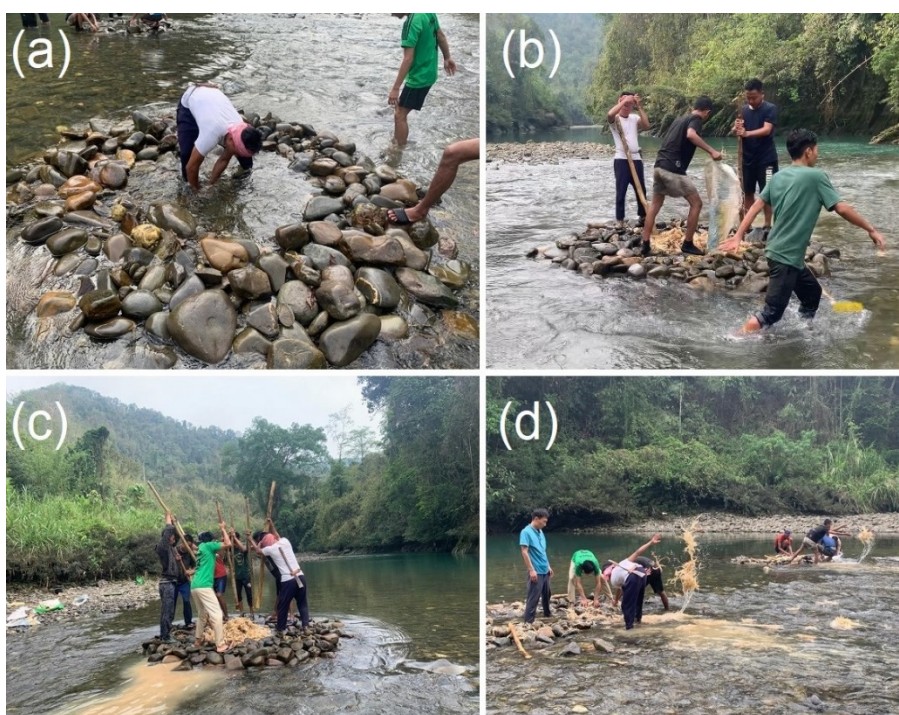

**Figure 8.** (**a**) Construction of stone platform for pounding and threshing of roots, (**b**) transfer of the roots of *D. elliptica* for pounding, (**c**) pounding of *D. elliptica* showing the release of yellowish extracts from the roots and (**d**) releasing the pounded root into the river current, during intra-tribal community fishing practices in Nagaland, Northeast India.

3.2.5. Other Activities and Structures Produced

Traditional Bamboo Barriers as Fish Traps

Temporary fish barriers, known as *Osa* in Lotha, are constructed to trap fish; this is the most widely used traditional barrier in Wokha and many parts of Nagaland. They are usually made of bamboo which is used wholly or split vertically and interwoven (Figure 9a). They are placed at the end point of the targeted stretch of river, which was also reported from Mizoram [31]. Water can easily pass through the interwoven bamboo; however, fish are trapped by the barrier which is manned by individuals from the villages. Sometimes the villagers simply install a cast net across the river which acts in a similar fashion by trapping the fish. The construction of the bamboo barriers was observed in both inter- and intra-community activities.

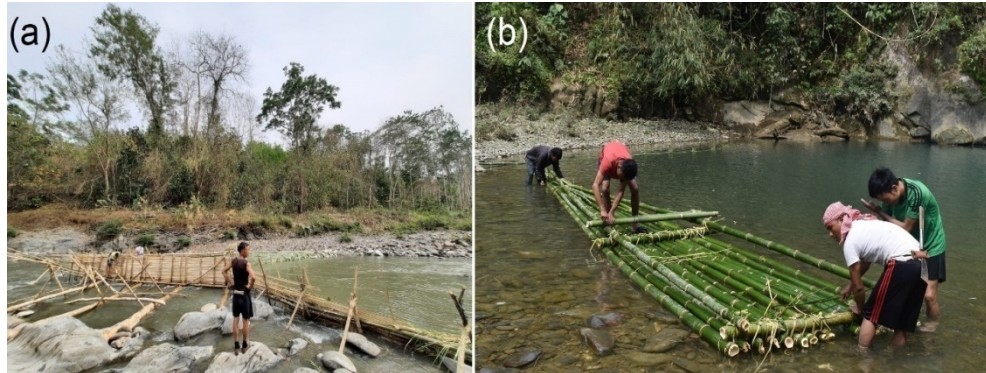

**Figure 9.** (**a**) Traditional bamboo fish barrier and (**b**) bamboo raft used for collecting fish after community fishing practices in Nagaland, Northeast India.

Construction of Temporary Bamboo Rafts

A temporary bamboo raft (Figure 9b) called *Yayüm* in Lotha is constructed prior to the start of harvesting. About 15 bamboo culms are fastened together and used as a raft for collecting fish from deeper sections of the water body which are dangerous and/or inaccessible. Construction of the bamboo rafts was also observed in both inter- and intra-community activities.

### 3.2.6. Harvesting and Distribution of Fish

Following the completion of pounding and release of the poison by the men participating in the activity, children, men and women are all involved in the collection of fish from the river 30 min after the pounding is finished with the harvesting of fish carried out throughout the rest of the day. Various fishing equipment such as cast nets, hooks and lines and other smaller nets are used during the harvest. Some of the most common fish caught are noted in Table 2 and illustrated in Figure 10a. In this type of community fishing, the fish caught by each individual are collected and stored clan-wise in a particular spot (Figure 10b) and distributed among the individuals of the clan following traditional values with the eldest person getting the biggest fish followed by the second largest to second eldest, this type of distribution practice is also reported in traditional fishing practices of the related Ao Naga tribe [32]. In the case of intra-community fishing observed among the Lotha tribes of the Chudi village, only men are present at the time of harvesting. The harvested fish are then collected in a specific location and distributed equally among the participants and other customary practices of distribution are followed such as reserving the largest fish for the oldest person in the village (as above).

**Table 2.** Common fish caught during community fishing events on the River Doyang, Nagaland, Northeast India.

| Scientific Name | Common Name |
| --- | --- |
| *Anguilla bengalensis* (Gray, 1831) (Anguillidae) | Indian mottled eel |
| *Sperata seenghala* (Sykes, 1839) (Bagridae) | Giant river catfish |
| *Channa* spp. (Channidae) | Common snakehead/Mudfish |
| *Cirrhinus mrigala* (Hamilton, 1822) (Cyprinidae) | Mrigal |
| *Labeo dyocheilus* (McClelland, 1839) (Cyprinidae) | Boalla |
| *Labeo pangusia* (Hamilton, 1822) (Cyprinidae) | Pangasia labeo/Naro |
| *Labeo rohita* (Hamilton, 1822) (Cyprinidae) | Indian major carp |
| *Tor tor* (Hamilton, 1822) (Cyprinidae) | Masheer |
| *Schistura manipurensis* (Chaudhuri, 1912) (Nemacheilidae) | Loach |

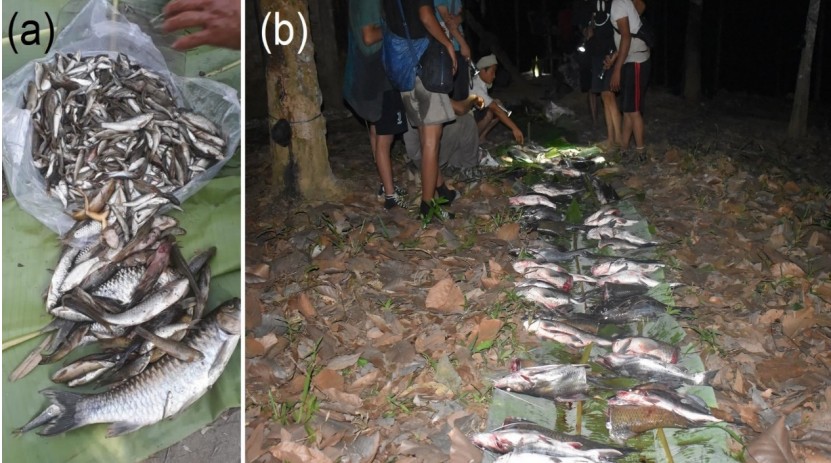

**Figure 10.** (**a**) Examples of fish caught and (**b**) distribution of fish after community fishing activities in Nagaland, Northeast India.

## 4. Discussion

We have described an important cultural phenomenon of community fishing practices within and between tribes of Nagaland showing a practice that sustains traditional ecological knowledge and cultural traditions. We observed two different types of indigenous fishing practice (inter- and intra-community) using two different piscicidal plants (*M. pachycarpa* and *D. elliptica*). The main motive of these inter-tribal and intra-tribal practices is to preserve and uphold the age-old tribal traditions and cultures while fostering peaceful co-existence and mutual prosperity within and among different communities/villages. This social solidarity is one the most important features of fishing communities at the local/village level as described by other studies particularly in the context of contemporary socioeconomic conditions [33]. There have been reports of inter-village/community conflicts between fisher folk [34,35] and therefore indigenous inter- and intra-community fishing practices are critical to bridging the gap between social, cultural and economic differences within/among the communities and promoting egalitarianism.

The Lotha-Sumi community fishing at Tsungiki and Philimi is one of the biggest fishing festivals in the state of Nagaland and the most successful traditional fishing activity in terms of participation in comparison to other community fishing activities. Although not quantified in this study, the harvesting of fish from this practice can be high, for example a similar study in Garhwal Himalaya recorded a 500 kg harvest (and up to 2 tons previously) [36] using a different piscicidal plant in a 'fishing festival' and a 200–300 kg or more harvest was reported from Mizoram [31]. It was also reported that some Naga communities/villages practice this traditional fishing every year [11] and some communities/villages carry out the operation once every two to four years depending upon their district/tribe and other customary laws, policies and regulations, and the frequency of the activities thus needs to be considered. For example, it was reported that Makú Indians in the Amazon practice poisoning in several year cycles [37]. How the removal of fish from the ecosystem is affecting its functioning has not been studied but it likely leads to a large effect. Few studies have shown the effects on non-target species and other aquatic biota [38–41] since the efficacy of the species of plant used as piscicides differs between fish species [42]; furthermore, it also reflects the negative impacts of the use of piscicides that have been outlawed in some other countries [43]. In addition, there is lack of research in relation to the magnitude of toxicity and longevity of the effects of the two plant species on both fish and the broader riverine ecosystem. On the other hand, the use of dynamite and synthetic piscicides such as calcium hydroxide Ca(OH)$_2$ and bleaching powder/calcium hypochlorite Ca(OCl)$_2$ are also increasingly utilized for fishing in Northeast Indian rivers [44,45], although research on this is difficult due to the illicit nature of their use. However, due to their non-biodegradable nature, unlike the organic piscicides, the synthetic residues stay in the water body for longer periods thus disrupting the riverine ecology [46] whilst possibly also bioaccumulating within the ecosystem [47]; therefore, these practices are not acceptable under the current scenario of ecosystem degradation and biodiversity loss. It should be noted that these synthetic pesticides are not used in the traditional community fishing practices described above. Information regarding the frequency of fishing using piscicides, synthetic fish poisons, electro- and dynamite fishing and their impact on the riverine ecosystems in the study area is unknown. It has been reported that the use of piscicidal plants is safer than synthetic alternatives since organic chemicals are more easily degraded in comparison to their synthetic counterparts [48]. Thus, botanical piscicides are believed to be more eco-friendly than synthetic piscicides. Knowledge on indigenous fishing practices is also important in the context of fisheries research and natural resource management. For example, a study from Nigeria concluded that indigenous knowledge can be used as a supplementary source and basis for new scientific investigation to obtain knowledge relating to the entire ecosystem [49]. Several other studies have also shown the importance of traditional knowledge in the field of fisheries research and management [50–53]. Additionally, there is a need to consider the sustainability of piscicidal plant harvest—the removal of roots is a more destructive practice than the removal of leaves and

other above-ground plant parts. This is important as we estimate that about 3 tons (fresh weight) of roots was used in the largest intra-community event organized and about 200 kg was used for the smaller inter-community fishing practice.

As noted by Lynch et al. [54], results from this work can contribute towards supporting the Sustainable Development Goals (SDGs): No Poverty (SDG 1), Zero Hunger (SDG 2), Clean Water and Sanitation (SDG 6), Responsible Consumption and Production (SDG 12) and Life on Land (SDG 15). For example, the majority of the tribal populations of the participating communities are mostly dependent on forest products, bush meat, fish and *jhum* agriculture for their sustenance but there is a lack of information on the impact of community fishing on the health or economic status of the participating individuals or communities. Diverse fish populations offer a healthy source of protein for the tribal population and can be an important source of livelihood for families [55]; however, chemical and dynamite fishing has overlapped eco-friendly traditional fishing methods exerting pressure on the tribal populations in relation to food security since these techniques are not economically available for the majority of the tribal inhabitants. Fishing practices such as fish farming with rice are efficient alternatives [56] and the development of fisheries in the Doyang Reservoir [23,24] can be considered further along with other aquacultural, small-scale and artisanal fisheries along the Doyang and Jüthümrhü rivers of the study area that can play a role in aiding food security and economic uplift of the tribal populations [45].

## 5. Conclusions

This study reflects the traditional values and practices, and highlights the events occurring during community fishing in Nagaland, Northeast India. Our study revealed indigenous fishing methods which showed a number of similarities but differed in their mode of practice and use of piscicidal plants depending on the groups involved. In the past few decades, fishing, hunting and agriculture practices were sustainable but stopping traditional practices including community fishing and *jhum* cultivation is near impossible since they are deeply rooted within the cultures, beliefs and customs of the tribal people. Therefore, there is a need for researchers and policy makers to determine and institute sustainable approaches without compromising the traditional practices. Considering the rapid loss of biodiversity in the highly diverse region of Northeast India, further research and development of management policies aimed at restoration, rejuvenation and conservation of riverine biodiversity are paramount. Firstly, it would be valuable to determine effective doses and impacts of specific piscicidal plants on the fish species' populations such that suitable recommendations can be made to allow rejuvenation and replenishment of the ecosystem as an approach to conservation and sustainability. Secondly, given the increasing use of synthetic non-biodegradable chemical compounds, the amendment of strict local/tribal policies and regulations related to indiscreet practices of fishing using these compounds should be considered. Thirdly, our study suggests that the use of piscicidal plants should be encouraged over synthetic compounds or other destructive fishing practices owing to the more eco-friendly nature of the poison and the cultural heritage associated with them, but it would be of value to determine the effects of harvesting a large amount of root material on the populations of these plants. Finally, the adoption of aquaculture, small-scale and artisanal fisheries to reduce pressure on the riverine ecosystems and other natural resources should be considered to support a number of SDGs. Our documentation of the traditional ethnic practices and traditional ecological knowledge is therefore vital under the current global scenario of biodiversity and cultural heritage loss.

**Author Contributions:** Conceptualization, E.Y.O., L.M.K., F.Q.B. and S.K.T.; investigation, E.Y.O.; writing—original draft preparation, E.Y.O., L.M.K., F.Q.B. and S.K.T.; writing—review and editing, E.Y.O., L.M.K., F.Q.B. and S.K.T. All authors have read and agreed to the published version of the manuscript.

**Funding:** This research received no external funding.

**Institutional Review Board Statement:** The institutional review board permit number allotted for this project is MZU-IAEC/2021/19 dated 26 March 2021.

**Informed Consent Statement:** All subjects gave their informed consent for inclusion before they participated in the study.

**Data Availability Statement:** Not applicable.

**Acknowledgments:** The authors would like to extend a heartfelt gratitude to the elders and village heads (*Gaon burah*) and chairmen of the Lotha and Sumi communities for sharing their invaluable knowledge and experience. We would also like to thank the community members participating during the fishing expeditions and their contributions to this endeavor.

**Conflicts of Interest:** The authors declare no conflict of interest.

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
