# Peer review of "Indigenous Community Fishing Practices in Nagaland, Eastern Indian Himalayas"

_sustainability, doi:10.3390/su14127049_

Round 1

Reviewer 1 Report

This manuscript was in good written. However, for the research journal, the objectives including the parameters involved must be stated clearly along with the research work. Then the findings are either significant or not significant easy to conclude. Should be carried out the recommendation output to strengthen the research elements.

Author Response

Please see the attachment below. Thank you!

Reviewer 2 Report

This is a very interesting manuscript about the little known practices of fishers in Nagaland, who use poisons from native plants to bring fish to the surface, where they are caught.  The story is nicely told, and I have only a few specific suggestions for improvement (see below).  I also have a few general suggestions. One is to adopt a more conventional organization of the manuscript, one that includes a Results section (the information in your Section 4).  Your Section 3 can be omitted, with the contents of the first paragraph moving to Section 4 and those of the second paragraph to the Introduction, where you first describe the plants and poisons.  You can also abbreviate the names of the two plant genera after the first usage, e.g. Derris elliptica for the first usage, thereafter D. elliptica.  For consistency, use fish in both the singular and plural (avoid fishes).  And then finally, it would be nice if you included more information about the synthetic compounds that are sometimes being used.  Just rotenone?  Or other types of poisons?  Also, do you know how the rituals change when synthetics rather than root extracts are used?  I should think that this would have major impacts on ritualistic and community activities.

Specific comments:

L16/17. Add a sentence at about this position, informing readers that fishing in this area relies on the use of plant-based poisons (this becomes clear later, but the flow would be better if the concept was introduced here).

L35. Triterpenoids

L36. Remove the word “organic.” — is clear that piscicidal plants are organic.

L50/68. I understand that little is known about these issues, but I note that there are no citations here.  Is this literally the case—no scientific publications at all?

L83. This is already stated, see L48/49.

L84. Delete Fig. 1 and refer to the maps in Fig. 2.  There are several later photos of the river, so no need to show it here.

L88. River is known. Can you provide a citation?

Fig. 2. It appears that the river divides in half toward the top of your map, flowing on in two separate directions, but at L84/87, you just describe one channel.  Please clarify.

Table 1.  Replace the less than 3,500 and less than 500 notation with ranges of the number of participants in both cases. 10 is less than 3,500, but 3,499 is also less than 3,500–so your notation is not very precise.

L113. Can you give the approximate number of elders and tribal leaders who participated?

L118/119. Earlier you introduced the Sumi and Lotha tribes, here you add the Wokha and Zunheboto, but the relationship is unclear.  For example, what is the relationship between the Lotha and the Wokha?  

L313. Among the individuals

L359. Populations offer

Conclusions section. This is well written with some real conclusions—nicely done.

Author Response

(The authors gave the same response as above.)

Reviewer 3 Report

The topic is timely and important; however, the manuscript is subject to the following comments.

Abstract: The abstract should be written in the context of background, objectives, methods, results, conclusions and policy implications.

The background of the study is poorly written in the context of energy and environment. See the following studies in this regard.

https://doi.org/10.1016/j.techsoc.2022.101963

https://doi.org/10.1016/j.scitotenv.2020.144183

Several statements are without references. Thoroughly see the manuscript and provide proper and latest references to support your arguments.

In my opinion, the literature section is very weak and needs careful consideration.

How can you compare and contrast your findings with other geographical regions of similar backgrounds?

The discussion section needs major improvements. The discussion should be inferred from research results and it should not be generic.

The authors have used many old references in their study. Avoid such old references and try to incorporate new ones, as significant work has already been done in these years.

The conclusion section is weak and needs room for improvement.

The authors should mention the limitations of the study at the end of the conclusion section.

Author Response

(The authors gave the same response as above.)

Round 2

Reviewer 1 Report

Good effort to improve all the comments. Hopefully, you can be more successful in the related research in the future.

Reviewer 3 Report

  1. The abstract is too large. It should be concise and convey the main theme of the article.
  2. The analysis section is still very weak. The authors should include statistical analysis to strengthen the study.  
  3. Some recent studies have never been incorporated, though they were suggested in the previous comments. 

Round 3

Reviewer 3 Report

This is very strange that the authors did not perform any analysis in their current work and selected its type as "Article".

This piece of work is just a report work that can be easily edited within no time. Scholarly articles are of high quality, which meet the criteria established by the esteemed Sustainability Journal. 

I don't think that the authors have paid attention to numerous comments conveyed in the previous rounds of revisions.

Author Response

Thank you for taking the time to review our manuscript.